# Tunable resistivity exponents in the metallic phase of epitaxial nickelates

Qikai Guo [1✉], Saeedeh Farokhipoor[1], César Magén[2,3], Francisco Rivadulla [4] & Beatriz Noheda [1,5✉]

We report a detailed analysis of the electrical resistivity exponent of thin films of $NdNiO_3$ as a function of epitaxial strain. Thin films under low strain conditions show a linear dependence of the resistivity versus temperature, consistent with a classical Fermi gas ruled by electron-phonon interactions. In addition, the apparent temperature exponent, $n$, can be tuned with the epitaxial strain between $n = 1$ and $n = 3$. We discuss the critical role played by quenched random disorder in the value of $n$. Our work shows that the assignment of Fermi/Non-Fermi liquid behaviour based on experimentally obtained resistivity exponents requires an in-depth analysis of the degree of disorder in the material.

[1] Zernike Institute for Advanced Materials, University of Groningen, 9747 AG Groningen, The Netherlands. [2] Instituto de Ciencia de Materiales de Aragón (ICMA) and Departamento de FÃsica de la Materia Condensada, Universidad de Zaragoza-CSIC, 50009 Zaragoza, Spain. [3] Laboratorio de Microscopías Avanzadas (LMA), Instituto de Nanociencia de Aragón (INA), Universidad de Zaragoza, 50018 Zaragoza, Spain. [4] CIQUS, Centro de Investigación en Química Biolóxica e Materiais Moleculares, and departamento de Química-Física, Universidade de Santiago de Compostela, Santiago de Compostela 15782, Spain. [5] CogniGron center, University of Groningen, 9747 AG Groningen, The Netherlands. ✉email: q.guo@rug.nl; b.noheda@rug.nl

The tunable resistivity of materials undergoing a metal–insulator transition (MIT) holds great promise for resistive switching applications, such as adaptable electronics and cognitive computing[1–7]. However, a complete understanding of the metallic phase in these strongly correlated electron systems is still one of the central open problems in condensed matter physics[8,9].

Electronic transport is generally explained by means of Boltzmann's theory, which considers a fluid of free quasi-particles that scatter occasionally. In normal metals, the resistivity increases linearly with temperature as electrons are more strongly scattered by lattice vibrations. At low temperatures, weak interactions between electrons can significantly affect the electrical properties, and give rise to a $T^2$ dependence of resistivity, according to Landau's Fermi liquid (FL) theory[10]. Therefore, the scaling exponent of the power-law term of the resistivity as a function of temperature ($n$) is often used to infer the type of interactions ruling the metal state. In materials with strong electron–electron interactions and undergoing ordering phenomena, other exponents ($n \neq 1$, 2) are usually observed, being the physics behind this so-called 'Non-Fermi liquid' (NFL) behaviour[11–13], a subject of active discussion[14–17].

Among strongly correlated electron materials, nickelates (RENiO$_3$, with RE denoting a trivalent rare-earth element) present a very interesting case. They have attracted attention due to their MIT[18] and the possibility to tune it using different RE elements or by epitaxial strain[19–24]. Bad metallic behaviour in nickelates has also been claimed[25]. Different models for the origin of the MIT have been put forward, based on either positive or negative charge transfer as responsible for the insulating state[26–34]. The negative charge transfer model supports the bond disproportionation picture, and is strongly supported by recent experiments[35–37]. Independent from the exact microscopic picture, the origin of the MIT is a cooperative lattice distortion that reduces the symmetry from a high-temperature orthorhombic phase to a low-temperature monoclinic phase, involving two Ni sites, with the associated need for cooperative accommodation of different Ni–O bond lengths[38]. Remarkably, it has been reported that eliminating the MIT in nickelates by orbital engineering would give rise to a superconducting state[39], with a very recent experimental achievement in this direction[40]. It becomes, then, important to have an accurate picture of the relevant electron interactions in the intermediate- and low-temperature regimes, just before the MIT takes place. However, despite the vast amount of recent works, the metallic behaviour of the nickelates is not yet fully understood.

In nickelates, different $n$ exponents of the resistivity as a function of temperature have been reported[14,25,41–46]. Linear dependence with temperature has been measured in the whole Nd$_x$La$_{1-x}$NiO$_3$ series in ceramic pellets[41]. Liu et al.[42] obtained $n = 5/3$ and $n = 4/3$ for NdNiO$_3$ (NNO) films under compressive strain, while Mikheev et al. reported a crossover between FL ($n = 2$) and NFL ($n = 5/3$) in NNO films with varying epitaxial strain[43]. The need for an empirical parallel resistor model to introduce the effect of the saturation resistivity rises questions about the interpretation of the apparent (experimentally obtained) exponents, as discussed by Hussey et al.[47].

Here, we report the evolution of the resistivity exponent of NdNiO$_3$ under different degrees of epitaxial strain. Strain-free (bulk-like) thin films show a linear temperature dependence of the resistivity ($n = 1$). The combined effect of epitaxial strain and random disorder produces a continuous departure from $n = 1$, in agreement with recent theoretical work by Patel et al.[48].

## Results

### Tuning the resistivity–temperature exponent in the metallic phase. 
Crystalline NNO films have been grown by pulsed laser deposition (PLD) on <001>-oriented LaAlO$_3$ (LAO), NdGaO$_3$ (NGO), SrTiO$_3$ (STO) substrates and <110>-oriented DyScO$_3$ (DSO) substrates, using a single-phase ceramic target (see the 'Methods' for more details). Perovskite NNO possesses an orthorhombic structure with a pseudocubic lattice parameter of 3.807 Å, which is slightly larger than that of the LAO substrate (3.790 Å). Thus, the films on LAO are expected to be subjected to small compressive strain. On the contrary, the films grown on NGO (3.858 Å), STO (3.905 Å) and DSO (3.955 Å) substrates should experience increasing tensile strain. Supplementary Fig. 1 (see Supplementary Note 1) shows the typical atomic force microscope (AFM) topography image of a 5-nm NNO film grown on a LAO substrate (NNO/LAO), showing that the atom-high steps from the substrate are still visible after the deposition of the film. In situ high-energy electron diffraction (RHEED) intensity oscillations recorded during the film growth indicate that at least the first 13 layers (~5 nm) of the NNO film are deposited atomic-by-atomic layer (see Supplementary Fig. 1a for NNO/LAO and NNO/STO films). The crystalline quality and strain state of the NNO films with different thickness and on different substrates was determined by X-ray diffraction (for details see Supplementary Note 1 and later discussions).

Figure 1a, b shows the sheet resistance ($R_S$) of NNO films grown on LAO and STO substrates, respectively, as a function of temperature. The NNO films grown on LAO substrates (under small compressive strain) exhibit a sharp MIT and a pronounced thermal hysteresis, while the hysteresis is strongly reduced in the NNO/STO films, in agreement with previous reports[1]. The evolution of the first-order transition towards a continuous, percolative-like metal–insulator transition is consistent with the presence of quenched random disorder in the films grown on STO[49]. This interpretation is supported by a higher resistivity and a smaller residual-resistivity ratio in these films compared with those grown on LAO. A further distinction is observed in the evolution of the metal–insulator transition temperature ($T_{MI}$) as a function of thickness (see insets to Fig. 1a, b), which has been attributed to the opposite alteration of orbital polarisation in response to different signs of the epitaxial strain[50].

Like in most of the metals, the electrical resistivity in the metallic state of nickelates can be fitted using a power law:

$$\rho(T) = \rho(0) + AT^n, \tag{1}$$

where $A$ is a coefficient related to the strength of electron scattering, and $n$ is the apparent power-law exponent. As shown in Fig. 1c, the metallic resistivity of all NNO films grown on LAO substrates in the measured temperature range (from $T_{MI} \sim$ 100–400 K) can be well described with a linear temperature dependence ($n = 1.00 \pm 0.01$), independent of film thickness. This temperature dependence has been observed in other systems, ranging from cuprates to heavy fermions, in spite of their different mechanisms of electron scattering[51]. What they have in common, however, is a constant scattering rate per kelvin ($\approx k_B/\hbar$), indicating that the excitations responsible for scattering are governed only by temperature. On the other hand, in the case of NNO/STO films (Fig. 1d), the temperature-resistivity scaling of films with different thickness deviates from linearity, showing the departure from this intrinsic mechanism. The values of $n$- and $A$ coefficients in both NNO/LAO and NNO/STO systems are shown, as a function of thickness, in Fig. 2a (for details on the determination of $n$, see Supplementary Note 2). Interestingly, $n$ shows a clear evolution with thickness in the NNO/STO films: $n$ decreases with increasing NNO/STO film thickness from a value of $n = 3.00 \pm 0.05$ for a 5-nm film to an apparent linear dependence ($n = 1.01 \pm 0.01$) for the thickest film (40 nm). To understand this behaviour, we turn to an in-depth structural characterisation of the films.

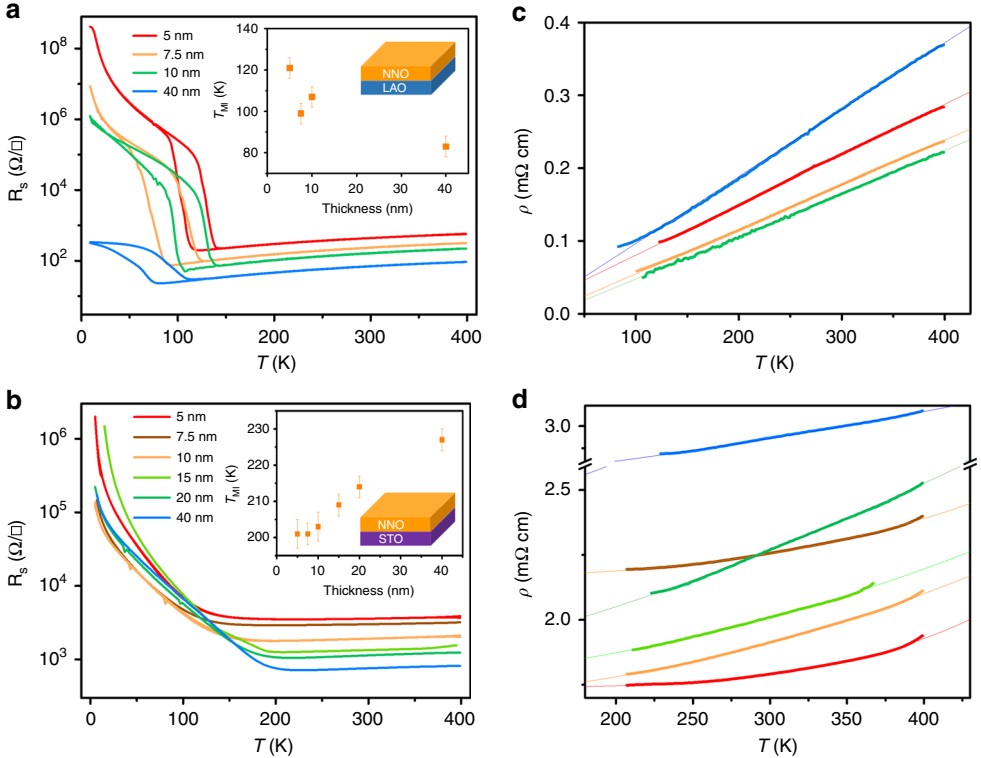

**Fig. 1 Temperature-dependent resistivity.** Temperature dependence of the sheet resistance ($R_S$), both during cooling and heating for NNO thin films grown on **a** LAO and **b** STO substrates with different thickness. The inset shows the metal–insulator transition temperature ($T_{MI}$) as a function of thickness. The $T_{MI}$ is extracted from the resistivity data during the cooling-down process. The resistivity as a function of temperature in the metallic phase of NNO thin films grown on **c** LAO and **d** STO substrates with different thickness. The thin solid lines are fits using Eq. (1).

Figure 3a, b shows the diffraction patterns for films grown on LAO and STO, respectively. The presence of Laue fringes indicates the high quality of the interfaces. The different sign of the epitaxial strain on the two substrates can be assessed by the different relative positions of the film and substrate peaks. Reciprocal space maps (RSM) around the $(103)_c$ peaks are shown in Fig. 3c–h. All NNO films grown on LAO grow coherently with the substrate (with coincident in-plane reciprocal lattices of film and substrate), for all investigated thicknesses, as expected from the very similar lattice of the bulk NNO (signalled in the maps by the yellow stars) and the substrate. On the contrary, in the NNO/ STO films, only the thinnest films grow coherently with the substrate, and show an in-plane lattice significantly larger than that of the bulk, due to the large differences between the bulk NNO and the STO substrate lattices. For increasing thicknesses, a gradual shift of the film peak can be observed, in agreement with the expected evolution of the lattice parameters and strain relaxation towards the bulk lattice, with increasing thickness. Thus, the observed evolution of $n$ (Fig. 2a) corresponds to the gradually relaxed in-plane strain of the films.

Figure 2b summarises the $n$ values extracted from the NNO films as a function of the in-plane strain, $\varepsilon_{xx}$, obtained from the diffraction data in Fig. 3. Data from NNO films on NGO substrates ($\varepsilon_{xx} = +1.34\%$) are also included. A 5-nm NNO/NGO film also shows apparent linear $T$ scaling in the metallic phase, confirming the correlation between the magnitude of the tensile strain and $n$ (see Supplementary Fig. 4). Similarly to the films on STO, the extended resistivity data of the NNO/NGO films (inset of Supplementary Fig. 4) also shows a reduced hysteresis compared with that of the films on LAO. Figure 2b is completed with $n$ values reported by other authors for bulk NNO[41] and NNO films under larger compressive strains[42,43]. Indeed, we

observe a clear dependence of $n$ on the in-plane strain. Both tensile and compressive strains are expected to induce an increase of the orbital splitting between the $Ni^{3+}x^2 - y^2$ and $3z^2 - r^2 e_g$ levels[43]. However, the large asymmetry observed, with a significantly stronger dependence for the tensile strain regime, points to an additional influence on $n$.

**Interplay between strain and defect formation.** In order to shed light into this behaviour, we performed scanning transmission electron microscopy (STEM) on the films. Cross-sectional specimens of the films were studied by atomic resolution STEM (for experimental details, see 'Methods'). The high-angle annular dark-field (HAADF) STEM image shown in Fig. 4a evidences the epitaxial, cube-on-cube growth of a 5-nm-thick NNO film on a LAO substrate, with a flat, atomically sharp interface. No defects or misfit dislocations are observed. The strain state of the films was determined by geometrical phase analysis (GPA) of the HAADF images; the deformation of the in-plane lattice parameter of the film with respect to the substrate ($\varepsilon_{xx}$) is depicted in Fig. 4b. $\varepsilon_{xx}$ is virtually zero across the 5-nm NNO film, showing a good in-plane lattice match between film and substrate, in agreement with the X-ray diffraction data. A thicker NNO film on LAO substrate also shows $\varepsilon_{xx} = 0$ across most of the film, but it starts showing small regions with Ruddlesden–Popper (RP) faults, often reported in nickelates[52], as seen in Fig. 4c, d. Some effect of these RP defects can be seen in the electrical properties, which show a strongly decreased resistance in the insulating state (Fig. 1a), as well as an increased resistivity in the metallic state for the 40-nm films on LAO (Fig. 1c). However, the PR defects do not preclude the presence of hysteresis at the metal–insulator transition, or the apparent linear behaviour of the metallic

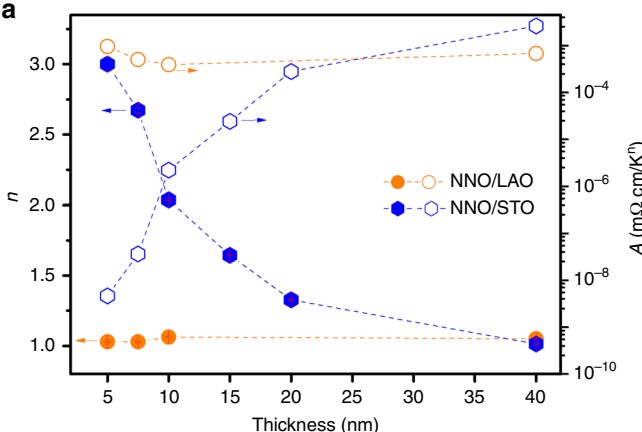

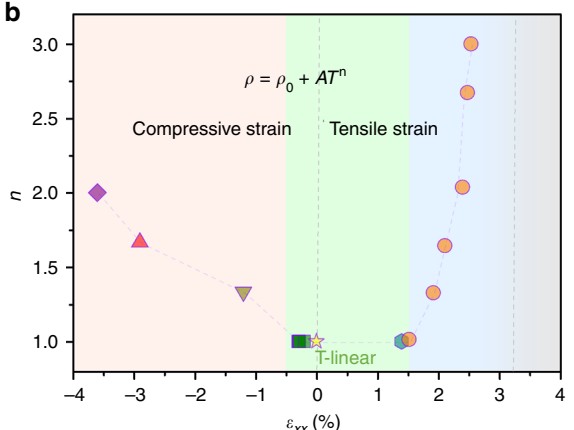

**Fig. 2 Tunable exponents. a** Power-law exponents ($n$) and $A$ coefficients from Eq. (1), extracted from the fits in Fig. 1 (c, d) as a function of film thickness. The error bars are determined as described in Supplementary Note 2. **b** Scaling exponent ($n$) as a function of in-plane strain ($\varepsilon_{xx}$). The data are for films grown on different substrates: LAO (squares), NGO (hexagon) and STO (circles). The highly tensile region shadowed in grey denotes the insulating state observed for the films on DSO. In addition, we also plot $n$ of bulk NNO[41] (star), as well as that for epitaxial NNO films under compressive strain reported by Liu et al.[42] (triangles) and Mikheev et al.[43] (rhombus).

resistivity in Fig. 1c, as it will be discussed in detail later. RP faults are known to have a significantly enlarged out-of-plane lattice parameter[52], which can explain the unusual evolution of the out-of-plane lattice parameters as a function of thickness for the NNO/LAO films, shown in Fig. 3a. Similar images for the thinnest and the thickest films on STO, shown in Fig. 5, reveal a higher abundance of RP faults, which are present even in the thinnest films. The data, thus, strongly suggest that the RP secondary phases present in the films are not correlated with the observed changes of $n$.

The effect of strain on $n$ may be indirect. Planar defects, such as misfit dislocations or stacking faults, have been often observed in nickelate films[53], and the creation of oxygen vacancies is known to be an efficient mechanism to relax tensile strain in epitaxially grown perovskites, as oxygen vacancies locally enlarge the lattice[22,24,54–57]. In nickelate-thin films, a pair of oxygen vacancies favour the reduction of the Ni ions to $Ni^{2+}$[6,58,59]. Indeed, measurement of Seebeck coefficients on films with a thickness of 10 nm grown on LAO and STO, shown in Fig. 6a, shows that while the film on LAO displays metallic-like transport, the film on STO shows a flat temperature dependence, a characteristic of polaronic systems.

Another indication of the existence of an increased content of oxygen vacancies in our films on STO comes from the structural data. From the definition of Poisson ratio, $v$, the pseudocubic lattice parameters that would correspond to the unstrained case for the different films can be estimated as $a_o = (2va+(1-v)c)/(1+v)$[60,61], where $a$ and $c$ are the in-plane and out-of-plane lattice parameters of the films, respectively, obtained from the structural data of Fig. 3, and $v = 0.30$ has been used for all films. The results, in Fig. 6b, show that the films on LAO display a lattice volume close to the bulk value, while the unit-cell volume of the films on STO is significantly increased, which is consistent with a larger oxygen vacancy content that decreases with increasing thickness. Moreover, the residual-resistivity ratio (RRR), which is often used as a measurement of materials' purity, increases with increasing thickness in the films on STO (Fig. 6c), also in agreement with a lower vacancy content in the thicker films.

Our experiments, therefore, indicate that NNO films subjected to relatively small strain values, display $T$-linear resistivity scaling. For larger values of tensile strain, an increase of the power- law resistivity–temperature exponent with the magnitude of the strain is observed. This is related to both the effect of strain on the orbital splitting and the degree of disorder, most likely due to oxygen vacancies, whose concentration is believed to increase with increasing tensile strain. These results validate recent theoretical predictions by Patel et al.[48]. Their computational work uses the Anderson–Hubbard Hamiltonian to predict that the metallic state that arises for small and intermediate values of both the on-site Coulomb interaction of $3d$ electrons ($U$) and the disorder ($V$) can be continuously tuned. The calculations predict values varying from $n = 1$ to $n = 2$ by the joint action of both $U$ and $V$ (it is to be noticed that in our experiments, larger values up to $n = 3$ are also observed). Interestingly, power-law exponents varying with the degree of disorder have also been reported for $SrRuO_3$ thin films by Herranz et al.[62].

## Discussion

In nickelates, epitaxial strain lifts the orbital degeneracy and causes orbital polarisation of the $e_g$ band: compressive strain lowers the energy of $3z^2 - r^2$ orbitals, while tensile strain lowers the $x^2 - y^2$ orbitals[43]. In this sense, both compressive and tensile strain have a similar influence on $U$. Since the amount of defects is smaller in the films under compressive strain, the values of $n$ under epitaxial compression should be a closer measure of the direct effect of strain in the absence of disorder. On the other hand, the introduction of oxygen vacancies in the tensile case gives rise to a combined effect of strain and disorder, which is reflected in a stronger dependence with strain in the tensile region of Fig. 2b. Actually, to directly clarify the effect of disorder on $n$, a plot of $n$ versus defect density, instead of epitaxial strain as in Fig. 2b, would be more appropriate. However, an accurate quantitative estimation of the amount of defects in such thin films is very challenging and could lead to erroneous conclusions (see Supplementary Note 4). Given the relationship between strain and defect concentration demonstrated by several authors[58,63,64], such a conservative plot is more adequate.

In addition, a direct investigation of the correlation between electrical transport properties and defect density can be achieved by tuning the concentration of oxygen vacancies of a single film by changing the annealing conditions after growth. For this, a 20-nm NNO film grown on a STO substrate with different amounts of oxygen vacancies was prepared in this work (see 'Methods' and Supplementary Note 5), and the corresponding changes in structure and resistivity were characterised (see Supplementary Fig. 6). As we mentioned above, the existence of oxygen vacancies gives rise to an enlarged unit-cell volume of the films. This is an

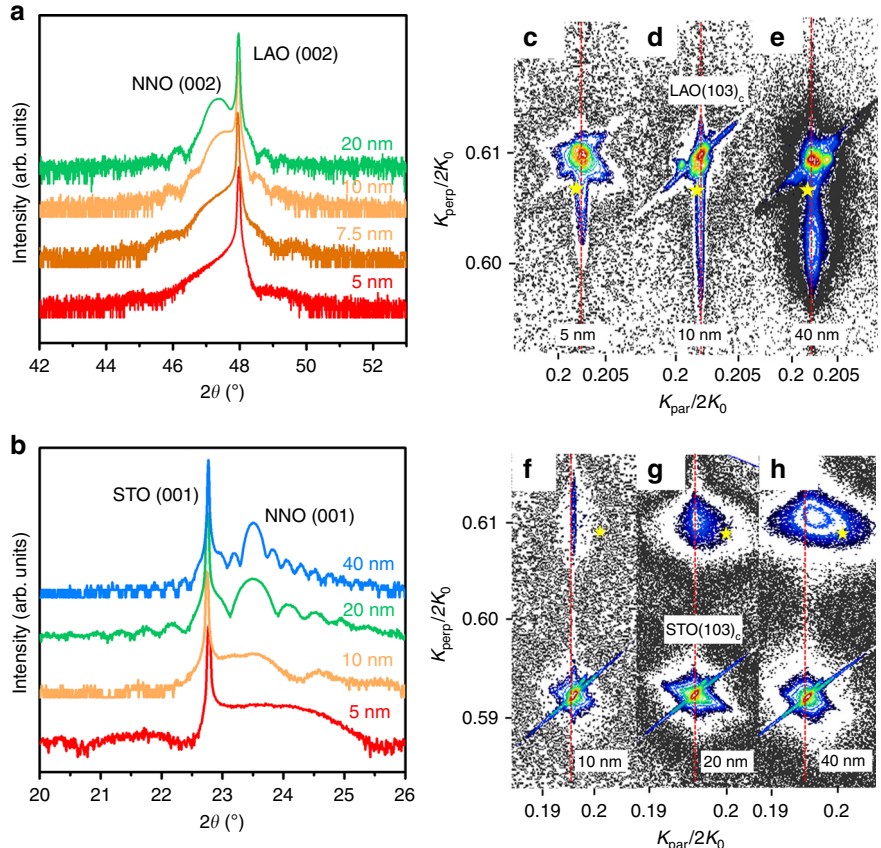

**Fig. 3 Structural characterisation.** X-ray diffraction patterns around the 002 reflection of NNO/LAO films (**a**), and around the 001 reflection of NNO/STO films (**b**) with different thicknesses. Reciprocal space map (RSM) around the (103)$_c$ diffraction peaks of **c** 5-nm, **d** 10-nm and **e** 40-nm NNO/LAO films, and **f** 10-nm, **g** 20-nm and **h** 40-nm NNO/STO films. The abscissa, $K_{perp}$, (ordinate, $K_{par}$,) represents the in-plane (out-of-plane) component of the scattering vector. Both are normalised by $2k_0 = 4\pi/\lambda$. The red dashed lines are guides to the eyes showing the substrate in-plane lattice. The yellow stars signal the (103)$_c$ peak of bulk NNO.

effect of chemical expansivity due to electrons being donated to $\sigma$ bands. Hence, the change in the density of oxygen vacancies is correlated with a change of the lattice parameters of the films[59]. As shown in Supplementary Fig. 6a, the out-of-plane lattice parameter of the 20-nm NNO/STO film after vacuum annealing is about 3.799 Å. This value is larger than 3.782 Å of the optimised film (see Fig. 3b), which has been annealed with a 900-mbar oxygen pressure, as explained in the 'Methods'. This is consistent with a larger content of oxygen vacancies for the vacuum-annealed films, as expected. As a consequence of this increase in oxygen vacancies, the metallic phase is fully suppressed, accompanied with several orders of magnitude increase in resistivity, as shown in Supplementary Fig. 6c. If the film is subsequently annealed in an oxygen-enriched environment at increasingly large temperatures, oxygen can be gradually replenished, resulting in a decrease of the out-of-plane lattice parameter and, thus, a shift of (002) diffraction peak towards larger angles. Correspondingly, the resistivity shows a decrease, and the metallic phase is recovered after annealing at sufficiently high temperature. More importantly, with the further reduction of oxygen vacancies, a clear evolution of the exponent $n$ from 2.24 to 1.64 is also observed in the resistivity of the metallic phase (see inset in Supplementary Figs. 6c and 7), deviating from the $T^{1.33}$ dependence measured for this thickness on samples annealed with the standard procedure (see Fig. 2a). For comparison, the same annealing treatment was also employed in a 20-nm NNO/LAO film. However, only a linear $T$ dependence of resistivity ($n = 1$) is found in this system after the recovery of the metallic phase,

regardless of the oxygen content (see Supplementary Fig. 6b, d). These experiments reveal that the oxygen vacancy content in the films on LAO is not large enough to induce changes in the macroscopic transport through the film, while the larger oxygen vacancy content in tensile-strained nickelate films clearly affects the resistivity–temperature-scaling exponent.

Next to vacuum annealing, a large enough tensile strain can also induce a large density of oxygen vacancies, and should, eventually, suppress the metallic phase. This is confirmed in films grown on DSO substrates, under +3.86% strain, for which the resistivity data can be described by a variable range hopping (VRH) conduction model for $T < 70$ K (see Supplementary Fig. 8) followed by a nearest-neighbour hopping (NNH) model with $E_a = 32$ meV for temperatures above $T = 70$ K, as often observed in disordered solids[65,66]. It is interesting to notice that a film of the same thickness on STO shows similar behaviour in the insulating state: comparable $E_a$ in the NNH regime and comparable crossover temperature from VRH to NNH conduction (see Supplementary Fig. 9). It is known that the presence of quenched disorder strongly impacts the transport properties inducing percolation and changing the nature of the phase transition[49]. In such percolation picture, a coexistence of metallic and insulating clusters could persist into the metallic phase. Indeed, the data of the films under intermediate strain (on STO) show a magnitude of the resistivity in the metallic state that is in between those of the film on DSO and the film on LAO. It is worth to point out that oxygen vacancies can also order in nickelates, as recently shown both in thin films[53] and bulk

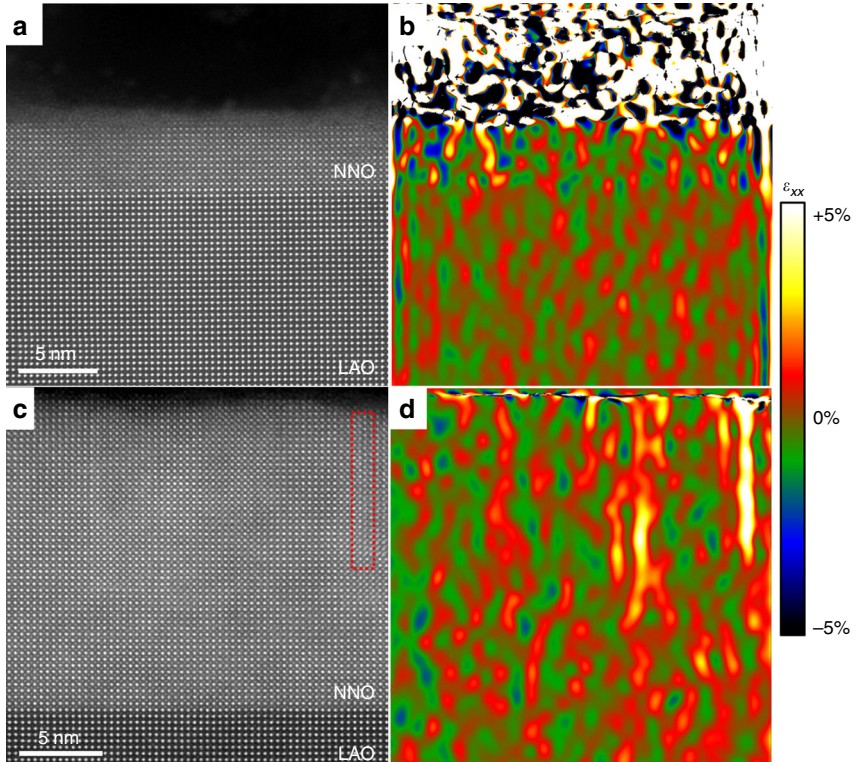

**Fig. 4 Atomic image and strain in NNO/LAO.** Cross-sectional HAADF-STEM image of NdNiO$_3$ thin films grown on LaAlO$_3$ substrates, for a 5-nm-thick film (**a**) and a 20-nm-thick film (**c**). The respective in-plane components of the strain tensor ($\varepsilon_{xx}$, colour scales) obtained from the STEM images by geometrical phase analysis (GPA) are shown in **b** and **d**. The red dashed lines surround the RP faults.

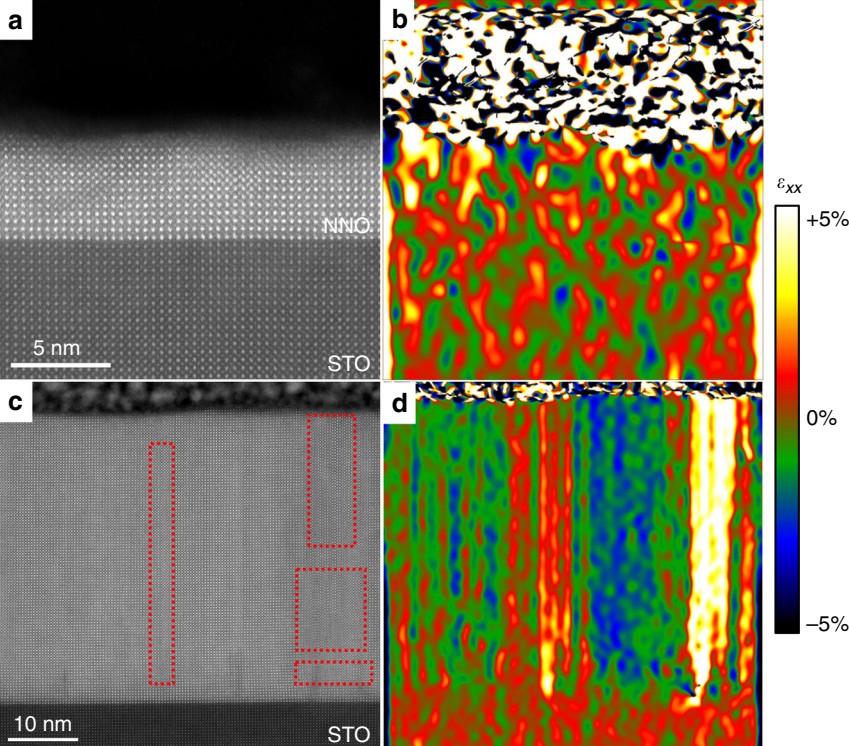

**Fig. 5 Atomic image and strain in NNO/STO.** Cross-sectional HAADF-STEM image of NdNiO$_3$ thin films grown on SrTiO$_3$ substrates, for a 5-nm-thick film (**a**) and a 40-nm-thick film (**c**). The respective in-plane components of the strain tensor ($\varepsilon_{xx}$, colour scales) obtained from the STEM images by geometrical phase analysis (GPA) are shown in **b** and **d**. The red dashed lines surround the RP faults.

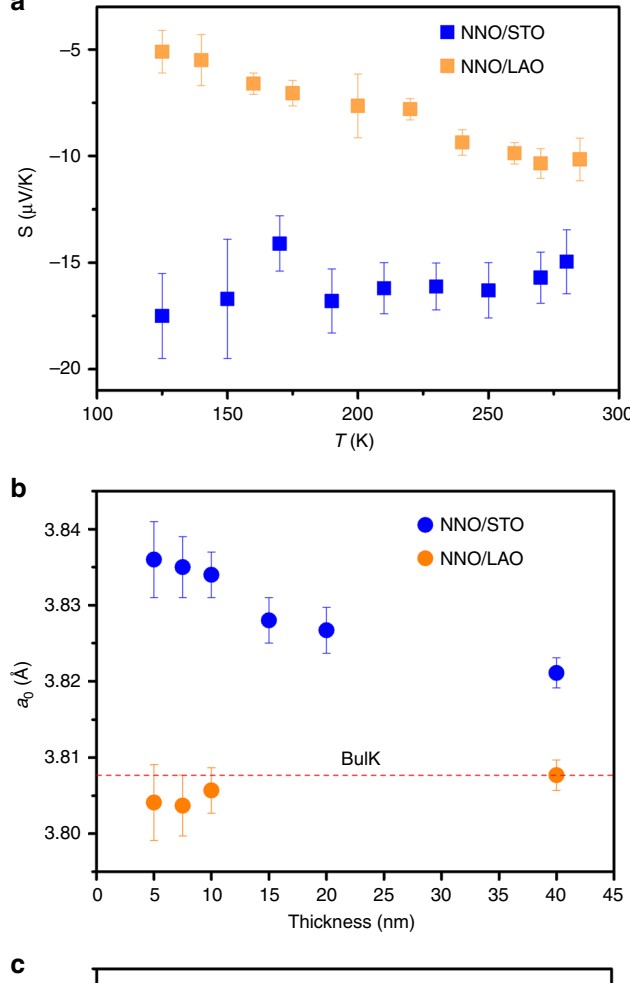

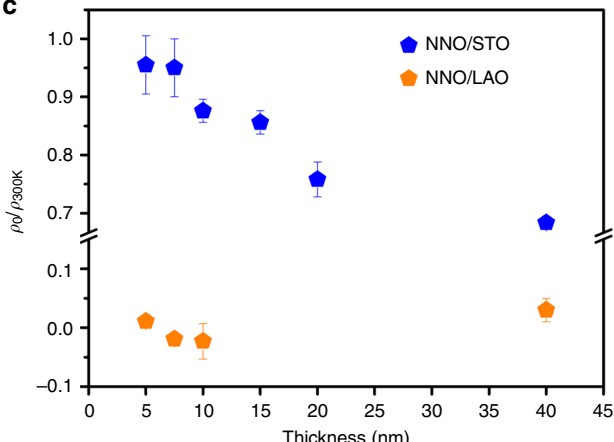

**Fig. 6 Oxygen vacancy indicators. a** Seebeck coefficients (S) measured on NNO films with thickness of 10 nm grown on LAO and STO substrates. **b** The unstrained film lattice parameter ($a_0$) and **c** the reversed residual-resistivity ratio ($\rho_0/\rho_{300K}$) of NNO films grown on LAO and STO substrates with different thickness. The error bar was determined from the deviation of repeated measurements.

crystals[67] of metallic $LaNiO_{3-\delta}$. The controlled tunability of oxygen vacancies with strain and its direct relationship with the transport properties demonstrated could also be of importance in the context of the bond disproportionation and negative charge transfer models[35], as well as the recent work proposing the metal state as a bipolaron liquid and the insulating phase as its ordered (bond-disproportionated) version[37].

To summarise, this work reports a clear evolution of the apparent scaling exponent of the resistivity–temperature characteristics ($n$) with strain and disorder, supporting recent theoretical predictions that show the tunability of the scaling exponents arising from the interplay between electron interactions and disorder in nickelates[48]. The overall picture helps to clarify that the underlying physics behind the observed evolution of exponents from $T$-linear to quadratic scaling and beyond, does not necessarily imply a crossover between FL and NFL behaviour or other exotic physics. On the contrary, for the films reported here with bulk-like in-plane lattice parameters, the contribution to the transport properties from delocalised electrons, for the intermediate-temperature region above the metal–insulator transition, is fully consistent with a classical Fermi gas ruled by electron–phonon scattering.

## Methods

**Materials' synthesis.** Epitaxial $NdNiO_3$ thin films were deposited on single-crystal $LaAlO_3$ (LAO), $NdGaO_3$ (NGO), $SrTiO_3$ (STO) and $DyScO_3$ (DSO) substrates by pulsed laser ablation of a single-phase target (Toshima Manufacturing Co., Ltd.). The quality of the target is of crucial importance to attain reproducibility of the film properties, as reported in ref. [68]. Before deposition, the LAO substrates were thermally annealed at 1050 °C in a flow of $O_2$ and etched with DI water to obtain an atomically flat surface with single terminated terraces. The NGO and STO substrates were etched with buffered $NH_4F$ (10 M)-HF solution (BHF), and the DSO substrates were etched with NaOH. All the substrates displayed single terminated terraces after the treatment. The substrates were heated to a temperature of 700 °C, prior to the deposition of the films, and were kept at that temperature during growth. Oxygen was present in the growth chamber during deposition with an oxygen pressure of 0.2 mbar, and the laser fluence on the target was 2 J/cm². After deposition, the samples were cooled down to room temperature at 5 °C/min with a oxygen pressure of 900 mbar. The growth was monitored using Reflection High Energy Electron Diffraction (RHEED). The films showed a constant deposition time of about 22 s per unit cell (s/uc) for NNO/LAO and 24 s/uc for NNO/STO. Films with various thicknesses were grown by precisely tuning the deposition time. The oxygen-deficient NNO films were grown on STO and LAO substrates followed by a vacuum-annealing process at $10^{-7}$ mbar. The concentration of oxygen vacancies in these films is tuned by annealing the specimens in tube furnace with a oxygen-enriched environment (400 cc/min) and step-by-step increased temperature. The annealing time for each step is 1 h.

**Structural characterisation.** The thicknesses, crystal orientation and phase purity of the films, as well as the epitaxial relation between the film and substrates, were assessed using X-ray diffraction by means of $2\theta-\omega$ scans and reciprocal space maps (RSM), respectively, on a Panalytical, Xpert MRD Pro diffractometer. Cross-sectional specimens of the films were prepared and studied by scanning transmission electron microscopy (STEM) on a probe-corrected FEI Titan 60–300 microscope equipped with a high-brightness field-emission gun (X-FEG) and a CEOS aberration corrector for the condenser system. This microscope was operated at 300 kV. High-angle annular dark-field (HAADF) STEM images were acquired with a convergence angle of 25 mrad and a probe size below 1 Å. The strain state of the films was determined by geometrical phase analysis (GPA) of these HAADF images.

**Electrical property measurement.** Electrical transport properties were measured between 5 K and 400 K by the van der Pauw method in a Quantum Design Physical Property Measurement System (PPMS), using a Keithley 237 current source and a Agilent 3458 A multimeter.

## Data availability

The data that support the findings of this study are available from the corresponding authors upon reasonable request.

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

## Acknowledgements

We are grateful to Manuel Bibes, Nigel Hussey and Jan Zaanen for insightful discussions that have helped reaching the current form of the paper. Useful discussions with Graeme Blake, Erik van Heumen, Arjun Joshua, Pavan Nukala and Mart Salverda are gratefully acknowledged. We are also indebted to Sanne Berg for the XPS measurements and analysis, as well as to Harry T. Jonkman and Ronnie A. Hoesktra for their help with the XPS analysis and the access to the facilities. In addition, we want to thank Jacob Baas and Henk Bonder for their invaluable technical support. Qikai Guo and Saeedeh Farokhipoor acknowledge financial support from a China Scholarship Council (CSC) grant and a VENI grant (016.veni.179.053) of the Netherlands Organisation for Scientific Research (NWO), respectively. Francisco Rivadulla acknowledges support by the Ministry of Science of Spain (Project No. MAT2016-80762-R), the Conselleria de Cultura, Educacion e Ordenacion Universitaria. Xunta de Galicia (ED431F 2016/008, and Centro singular de investigación de Galicia accreditation 2016–2019, ED431G/09), the European Union (European Regional Development Fund (ERDF)) and the European Commission through the project 734187- SPI-COLOST (H2020-MSCA-RISE-2016).

## Author contributions

Q.G. and B.N. designed the experiments. Q.G. grew and characterised the NNO films with help from S.F. Q.G. analysed the structure and transport of the data under the supervision of B.N. C.M. performed the STEM characterisation and analysis. Seebeck coefficients were measured and analysed by F.R. All the authors discussed the results. Q.G. and B.N. wrote the paper with contributions from all authors.

## Competing interests

The authors declare no competing interests.
