## [Peer Review File · Nature Communications]

Reviewers' comments:

Reviewer #1 (Remarks to the Author):

Report corresponding to the manuscript NCOMMS-19-30498:

Tunable resistivity exponents in the metallic phase of epitaxial nickelates

By Qikai Guo *et al.*

In this paper, the authors investigate the origin of the temperature-dependence of the resistivity of NdNiO₃ (NNO) thin films as function of thickness. Particularly, they focus on the evolution of the electrical resistivity exponent n ($\rho \propto AT^n$) for NNO films grown under different strain conditions (and different thickness). Whereas NNO films grown under small compressive strain (NNO/LAO) show an exponent $n=1$ (indicating linear dependence of the temperature versus resistivity), n reaches 3 as epitaxial strain increases (NNO/STO). For these NNO/STO samples, the exponent n is also found to change with film thickness. This last result is explained by the presence of tensile strain, which influences orbital polarization, and due to disorder, mainly through oxygen vacancies, and, thus, there is no need to invoke a crossover from a Fermi- to a Non-Fermi liquid behaviour to explain the evolution of n with strain.

The study of the nature of the MIT in nickelates in general, and thin films in particular, has been especially intense in the last decade. The analysis of the results is detailed and supported by a large variety of characterization/measuring techniques (including x-ray diffraction, AFM, TEM, transport measurement and Seebeck not). Nevertheless, most of the ideas presented here are not new or base on films that it is not clear to what extent represent the intrinsic behaviour of strained nickelate films, as detailed in the following:

- The fact that tensile strain favours the formation of oxygen vacancies in oxide films, it is not a new finding but rather a well-established fact in the oxide community (see for instance *Nano Lett.* 17, 794 (2017)) and also in nickelates (see for instance *Appl. Phys. Lett.* 91, 192110 (2007); *APL Mater.* 2, 116110 (2014)).

Did the authors try to anneal the films? Did they optimize the growth conditions for each strain value?

- The NNO/STO films investigated in the present paper do not show hysteresis. Moreover, the authors claim that the full suppression of the hysteresis in those samples is in agreement with previous results. I disagree on that: see for instance Ref. *Nat. Commun.* 4, 2714 (2013) or even the paper the authors themselves cite as an example of such behaviour (see Fig 2c of *Adv. Mater.* 22, 5517 (2010)). The resistivity values of the NNO/STO films also seem higher in the present manuscript than compared to previous works.

Additionally, in the paper it is mentioned that NNO films grown on NGO “show a non-hysteretic phase transition similar to that of NNO/STO” This statement is again in contradiction with previous published data, and even with authors own data – see inset Fig. S4.

- In addition to NNO/STO, this manuscript results also base on NNO/LAO films. In that case, it had been previously shown that NNO/LAO films may exhibit a MIT or metallic character (*i.e.*, *Nat. Commun.* 4, 2714 (2013); *Appl. Phys. Lett.* 106, 092104 (2015)), a behaviour which has sometimes been attributed to vacancy formation. How this would match the picture presented in the paper?

- The presence of RP faults also suggests that film growth could eventually be further optimized. Despite the authors say that RP do not influence the observed n -evolution, their presence also brings the question how representative are those films of intrinsic strained NNO films properties.

Additional comments,

- NNO/STO films: the exponent evolves from $n=3$ to $n=1$ as the thickness increase. In addition to disorder and strain, how is the role of other interface phenomena considered (i.e. polar discontinuity, oxygen octahedral coupling between orthorhombic and cubic structures, intermixing, etc.)?

- In the text, it is mentioned that NNO/LAO grows under “small compressive strain”, as one can derive from the lattice parameter difference of both compounds. However, in the abstract is mentioned “strain free”. What is the conclusion?

By the way, given that one wants to study the evolution of resistivity as function of strain, the ξ strain values should be indicated.

- Fig. 1a: How do the authors explain the shift in tendency of the 7.5 and 10nm NNO/LAO films? Has this result been reproduced?

Minor comments:

- Mention substrate orientation, especially for orthorhombic substrates.
- Indicate how ξ_{xx} strain is calculated.
- X-ray diffraction pattern of the 40nm NNO/LAO film: I believe there is a typo because it looks much thinner.
- Reference titles: ndnio3 should be replaced by NdNiO3

Because of all that, I do not recommend the publication in Nature Communications.

Reviewer #2 (Remarks to the Author):

This paper reports on the mechanism of the metallic conduction in perovskite nickelates as investigated by the analysis of the exponent of temperature-dependence of the resistivity. The conduction mechanism in nickelates is a long-standing issue of controversy. The authors prepared thin films of nickelate under different degrees of epitaxial strain. The resistivity of the film under small epitaxial strain shows T-linear dependence, reflecting the Fermi-liquid-like conduction. As increasing the epitaxial strain in the films, the temperature exponent becomes larger, indicating that the transport mechanism changes into non-Fermi-liquid-like conduction by the strain. The authors have investigated how the strain affects the carrier transport in the films by employing the TEM observation. It revealed that the quenched disorder induced by the strain dominates the change in the transport mechanism rather than the modification of the orbital degeneracy. The data shown in this paper is very clear and the deduced conclusion seems to be reasonable. This paper may contribute to converging the divergent opinions on the transport mechanism of metallic nickelate. The referee considers that the conclusion of this paper will be more strengthened by plotting the temperature exponent the resistivity as a function of the defect density rather than the strength of the epitaxial strain (Fig. 2b).

Reviewer #3 (Remarks to the Author):

Nickelates have been attracting a great attention especially after the realization of the superconductivity in infinite layer nickelates. The authors timely provide deeper understanding of the MIT in nickelates and are important to the rapid growing field. After going through the manuscript in detail, I found couples of scientific questions which need to be addressed before I recommend the publication of the manuscript in Nature Communications.

First of all, a lot of structure evidence is shown for the deviation of the scaling exponent n from $n = 1$ and most of the discussion is also focused on tensile strain region, while I found very little discussion on the compressive part. What is the mechanism responding for compressive part should be explicitly addressed.

The second comment is why the TMIT decreases with increasing thickness is not explicitly addressed. It is expected to see that the TMIT will move forward to bulk like for thicker films due to strain relaxation, which indicates the TMIT should increase with increasing thickness for NNO/LAO. I would agree with the explanation for NNO/NGO where the strain relaxation picture well stands with the observed results. For ultrathin film, the interfacial modulation of the octahedral tilt modified by substrates should play a role, since the recent report does show the import role of interfacial oxygen octahedral coupling for nickelate when nickelate is very thin. Third and more technically, how to define TMIT is not clear, especially for MIT with large hysteresis, such as TMIT for 5 nm NNO/LAO.

The last comment is that is there direct evidence to show the presence of the Ni²⁺ in tensile nickelate, such as EELS or XAS? This will strongly validate the role of oxygen vacancies and the proposed model.

Dear Editor,

We are grateful to the reviewers for critically reading our manuscript and for their pertinent comments and suggestions. In the following we answer all their comments one by one.

Reviewer #1.

In this paper, the authors investigate the origin of the temperature-dependence of the resistivity of NdNiO₃ (NNO) thin films as function of thickness. Particularly, they focus on the evolution of the electrical resistivity exponent n ($\rho \propto AT^n$) for NNO films grown under different strain conditions (and different thickness). Whereas NNO films grown under small compressive strain (NNO/LAO) show an exponent $n=1$ (indicating linear dependence of the temperature versus resistivity), n reaches 3 as epitaxial strain increases (NNO/STO). For these NNO/STO samples, the exponent n is also found to change with film thickness. This last result is explained by the presence of tensile strain, which influences orbital polarization, and due to disorder, mainly through oxygen vacancies, and, thus, there is no need to invoke a crossover from a Fermi- to a Non-Fermi liquid behaviour to explain the evolution of n with strain. The study of the nature of the MIT in nickelates in general, and thin films in particular, has been especially intense in the last decade. The analysis of the results is detailed and supported by a large variety of characterization/measuring techniques (including x-ray diffraction, AFM, TEM, transport measurement and Seebeck). Nevertheless, most of the ideas presented here are not new or base on films that it is not clear to what extent represent the intrinsic behaviour of strained nickelate films, as detailed in the following:

-Q1. The fact that tensile strain favours the formation of oxygen vacancies in oxide films, it is not a new finding but rather a well-established fact in the oxide community (see for instance Nano Lett. 17, 794 (2017)) and also in nickelates (see for instance Appl. Phys. Lett. 91, 192110 (2007); APL Mater. 2, 116110 (2014)).

We agree with the referee that the relationship between tensile strain and the increase of oxygen vacancies is well established, with many papers on the subject (see refs. 22, 24 and 53 in the first version of the manuscript). We now add the papers mentioned by the referee (see refs. 54- 56 in the new version).

-Q2. Did the authors try to anneal the films? Did they optimize the growth conditions for each strain value?

As with other materials grown in our lab, every paper is preceded by an intense materials optimization process, which requires months of work. This process also includes various experiments to optimize the oxygen content in the films as follows:

a) In the process we learned that the target was crucial to obtain reproducible growth (as mentioned in the paper) and, since then, we have grown NdNiO₃ samples on LaAlO₃, SrTiO₃ and other substrates using a single-phase target. From Figure 1, it is clear that the target obtained from the company Toshima, (see [1]) gives rise to a purer material than that of a homemade (stoichiometric but not single phase) target. Most films grown using the single-phase target displayed reproducible results in terms of lattice parameters and resistivity as a function of temperature (including reproducible hysteresis values), with consistent trend as a function of thickness.

Figure 1. Powder XRD of homemade NdNiO₃ target (black) and the target purchased from Toshima (red). The vertical red ticks on the bottom indicate the room temperature peak position of single crystalline NdNiO₃ with lattice parameters from Ref[2].

b) The optimized thin film growth pressure is $P_{O_2} = 0.2 \text{ mbar}$. Changes in the P_{O_2} during growth give rise to clear resistivity changes, as the referee can see in Figure 2. Except for the film grown under extremely low oxygen pressure (0.02 mbar), other films show a quite comparable resistivity in the metallic phase. However, the growth pressure can indeed give rise to a drastic change in the transition temperature, hysteresis and resistance in the insulating state. Among them, the films grown with $P_{O_2} = 0.2 \text{ mbar}$ show the sharpest metal-insulator transitions (MIT). This optimal pressure is in agreement with other literature reports [1,3].

Figure 2. Electrical resistivity as a function of temperature of NdNiO_3 films with thicknesses of 5 nm grown on LaAlO_3 substrate under different oxygen pressure.

c) In addition, the films are annealed inside the growth chamber immediately after the growth. As shown in Figure 3, films annealed under vacuum show a much larger resistivity than those annealed with oxygen and the metallic phase is even absent in these vacuum-annealed films below 400K, indicating a larger amount of oxygen vacancies in the lattice. For films annealed in oxygen atmosphere, the resistivity shows a sharper MIT for films annealed under a higher oxygen pressure. The optimized specimens were cooled down to room temperature at a rate of $-5 \text{ }^\circ\text{C}/\text{min}$ under a high oxygen pressure of 900 mbar (this is the maximum pressure we can apply inside the growth chamber). All films mentioned in the manuscript are annealed under these conditions after growth.

As it is well known to the community [4], oxygen vacancies have an opposite effect on the resistivity in the metallic and the insulating phase (they increase the resistivity in the former and they decrease it in the latter). Moreover, the reduction of annealing pressure exerts a more obvious effect on the resistivity of the NNO/STO films than in that of the NNO/LAO films. In other words, the NNO films grown on STO substrates are more sensitive to an oxygen-deficient environment than those on LAO, which is consistent with a higher tendency to the formation of oxygen vacancies under tensile strain [5].

Figure 3. Resistivity as a function of temperature of films grown on (a) LAO and (b) STO substrates under three different annealing processes.

The NNO films grown on LAO using the optimized conditions mentioned above show a very sharp MIT and display record resistance ratios of 6 orders of magnitude between the metallic and the insulating states. To our best knowledge, this ratio is larger than those reported in the literature [6-9], which indicates a very pure phase. For this work, after optimization of the films on LAO, the growth conditions are kept the same for all films on STO, NGO, and DSO substrates. In this way, we can claim that the observed difference in oxygen content of different films arises from the different strain conditions rather than the growth and annealing process.

References:

- [1] Preziosi, Daniele, et al. "Reproducibility and off-stoichiometry issues in nickelate thin films grown by pulsed laser deposition." *AIP Advances* 7.1 (2017): 015210.
- [2] Lacorre, Ph, et al. "Synthesis, crystal structure, and properties of metallic PrNiO₃: Comparison with metallic NdNiO₃ and semiconducting SmNiO₃." *Journal of Solid State Chemistry* 91.2 (1991): 225-237.
- [3] Preziosi, Daniele, et al. "Direct Mapping of Phase Separation across the Metal–Insulator Transition of NdNiO₃." *Nano letters* 18.4 (2018): 2226-2232.

- [4] Nikulin, I. V., et al. "Oxygen nonstoichiometry of NdNiO_{3-δ} and SmNiO_{3-δ}." *Materials research bulletin* 39.6 (2004): 775-791.
- [5] Conchon, Florine, et al. "Investigation of strain relaxation mechanisms and transport properties in epitaxial SmNiO₃ films." *Journal of Applied Physics* 103.12 (2008): 123501.
- [6] Mikheev, Evgeny, et al. "Tuning bad metal and non-Fermi liquid behavior in a Mott material: Rare-earth nickelate thin films." *Science advances* 1.10 (2015): e1500797.
- [7] Scherwitzl, Raoul, et al. "Electric-Field Control of the Metal-Insulator Transition in Ultrathin NdNiO₃ Films." *Advanced Materials* 22.48 (2010): 5517-5520.
- [8] Hauser, Adam J., et al. "Temperature-dependence of the Hall coefficient of NdNiO₃ thin films." *Applied Physics Letters* 103.18 (2013): 182105.
- [9] Preziosi, Daniele, et al. "Direct Mapping of Phase Separation across the Metal-Insulator Transition of NdNiO₃." *Nano letters* 18.4 (2018): 2226-2232.

-Q3. The NNO/STO films investigated in the present paper do not show hysteresis. Moreover, the authors claim that the full suppression of the hysteresis in those samples is in agreement with previous results. I disagree on that: see for instance Ref. Nat. Commun. 4, 2714 (2013) or even the paper the authors themselves cite as an example of such behaviour (see Fig 2c of Adv. Mater. 22, 5517 (2010)). The resistivity values of the NNO/STO films also seem higher in the present manuscript than compared to previous works.

We agree with the referee that papers mentioned do show hysteresis on the film on STO. A closer look at our own film (in figure 4) also shows that the hysteresis is very small but present. Thus, even if the hysteresis is clearly smaller in the films on STO than in the films on LAO, it is not fully suppressed. We have, thus, removed the sentence:

“the full suppression of the hysteresis in those samples is in agreement with previous results”.

Figure 4. Close-up of the resistivity versus temperature curve for a 10 nm thick NNO film on STO, showing that the hysteresis, even if clearly smaller than for the samples on LAO, has not fully disappeared.

Indeed, the resistivity of our NNO/STO films is a somewhat higher than in previous works, as correctly pointed out by the referee. We have compared the resistivity of our 5 nm film with those having a comparable thickness in previous works, as shown in Figure 5.

We agree with the referee that the higher resistivity in our case indicates more oxygen vacancies compared to other NNO/STO films. This is related to the answer to the previous point (Q2), where we explain that the growth conditions are optimized for the films grown on LAO and kept unchanged for different substrates in order to be able to extract the effect of strain alone. Optimization of the oxygen content in the films on STO to obtain the lowest resistivity would have required a higher oxygen pressure during growth or larger annealing times. However, by doing that, one runs the risk of modifying other parameters in uncontrolled ways. For example, changing the P_{O_2} changes the kinetic energy of the ions arriving at the substrate surface. We, thus, stand by our conclusions that the change in strain induces quenched disorder due to a larger presence of oxygen vacancies in the film and that this affects the resistivity exponents.

Recently, several works reported that rare-earth nickelates show non-Fermi liquid behavior and even a crossover between Fermi liquid to non-Fermi liquid

based on their experimentally obtained resistivity exponents [3-5]. We believe our results are important because they reveal that resistivity exponents of nickelates are very sensitive to the degree of disorder in the material. In this sense, it is not reliable to classify nickelates as Fermi liquid or non-Fermi liquid systems based on the experimentally obtained resistivity exponents without an in-depth analysis of the disorder in the material.

Figure 5. Comparison of the resistivity of our 5 nm thick films on STO with those of references [1] and [2] for the same thicknesses.

Reference:

[1] Liu, Jian, et al. Strain-mediated metal-insulator transition in epitaxial ultrathin films of NdNiO₃. *Applied Physics Letters* 96.23 (2010): 233110.

[2] Liu, Jian, et al. Heterointerface engineered electronic and magnetic phases of NdNiO₃ thin films. *Nature communications* 4 (2013): 2714.

[3] Mikheev, Evgeny, et al. Tuning bad metal and non-Fermi liquid behavior in a Mott material: Rare-earth nickelate thin films. *Science advances* 1.10 (2015): e1500797.

[4] Yadav, Ekta, et al. Influence of Cu doping and thickness on non-Fermi liquid behaviour and metallic conductance in epitaxial PrNiO₃ thin films. *Applied Physics A* 124.9 (2018): 614.

[5] Middey, S., et al. Epitaxial strain modulated electronic properties of interface controlled nickelate superlattices. *Physical Review B* 98.4 (2018): 045115.

-Q4. Additionally, in the paper it is mentioned that NNO films grown on NGO “show a non-hysteretic phase transition similar to that of NNO/STO” This statement is again in contradiction with previous published data, and even with authors own data – see inset Fig. S4.

We again agree with the referee that hysteresis is present in our NNO/NGO films. We have changed the sentence “ show a non-hysteretic phase transition similar to that of NNO/STO ” by “shows a reduced hysteresis compared to that of NNO/LAO ”

-Q5. In addition to NNO/STO, this manuscript results also base on NNO/LAO films. In that case, it had been previously shown that NNO/LAO films may exhibit a MIT or metallic character (i.e, Nat. Commun. 4, 2714 (2013); Appl. Phys. Lett. 106, 092104 (2015)), a behaviour which has sometimes been attributed to vacancy formation. How this would match the picture presented in the paper?

The results shown in these mentioned papers do not fully agree with our own results. We agree that the suppressed MIT in nickelates films is usually attributed to the low growth temperature [1] or vacancy formation [2,3]. For example, Adam et al.[1] observed a robust MIT in NNO/LAO film grown under 300 mTorr while it is completely suppressed in those grown under 9 mTorr, due to a smaller oxygen content in the film (it is accepted that the robustness of the MIT can be used as an indicator of optimized growth of film). In our films on LAO we have always found a robust MIT transition for thicknesses above 3 nm. For instance, the 5 nm NNO/LAO films displays a record resistance ratio of 6 orders of magnitude between the metallic and the insulating states.

The film quality and vacancy formation can also be assessed by comparing resistivity data with the works mentioned by referee, shown in Figure 6. Although the influence of vacancy formation on the metallic part of the resistivity of NNO/LAO films is not obvious, as revealed in [3] and also in Figure 2, it can be clearly seen in Figure 6, with the resistivity of our NNO/LAO film being smaller than those of the films in refs. [2] and [3] (digitalization software has been used to extract the data of refs. [2] and [3]; we apologize for the noisy quality of data in ref.[3], which does not make justice to the original data.)

Figure 6. Comparison of the resistivity of our 5 nm thick films on LAO with those of references [2] and [3] for the same thicknesses.

References:

- [1] Kaur, Davinder, J. Jesudasan, and P. Raychaudhuri. "Pulsed laser deposition of NdNiO₃ thin films." *Solid state communications* 136.6 (2005): 369-374.
- [2] Liu, Jian, et al. "Heterointerface engineered electronic and magnetic phases of NdNiO₃ thin films." *Nature communications* 4 (2013): 2714.
- [3] Hauser, Adam J., et al. "Correlation between stoichiometry, strain, and metal-insulator transitions of NdNiO₃ films." *Applied Physics Letters* 106.9 (2015): 092104.

-Q6. The presence of RP faults also suggests that film growth could eventually be further optimized. Despite the authors say that RP do not influence the observed n-evolution, their presence also brings the question how representative are those films of intrinsic strained NNO films properties.

We understand the concerns of the referee. We are confident that our films on LAO are optimized (see previous explanations). Indeed, detailed TEM analysis show that the 5 nm thick films contain no defects. The RP faults appear at

larger thicknesses (rarely for 10 nm thick films and more often for 20 nm and thicker films). Nevertheless, this increase of the volume ratio of the RP phase is not accompanied with a change in the resistivity exponent n . Therefore, we conclude that the RP secondary phases present in the films are not correlated with the observed changes of n .

It is worth to notice that most papers do not show a detailed quantitative analysis of the presence of this type of defects. However, given their abundance in transition-metal perovskites, including nickelates [1-3], it can be assumed that they are present also in other films in the literature. From this point of view, our films are representative of the intrinsic behavior of strained NNO films (including their natural presence of RP defects at increasing thickness).

References:

[1] Detemple, E., et al. "Ruddlesden-Popper faults in LaNiO₃/LaAlO₃ superlattices." *Journal of Applied Physics* 112.1 (2012): 013509.

[2] Coll, Catalina, et al. "Simulation of STEM-HAADF Image Contrast of Ruddlesden–Popper Faulted LaNiO₃ Thin Films." *The Journal of Physical Chemistry C* 121.17 (2017): 9300-9304.

[3] Seol, Daehee, et al. "Flexoelectric healing of intrinsically more conductive nanochannels in NdNiO₃ thin films." *Applied Surface Science* 497 (2019): 143727.

Additional comments,

-Q7. *NNO/STO films: the exponent evolves from $n=3$ to $n=1$ as the thickness increase. In addition to disorder and strain, how is the role of other interface phenomena considered (i.e. polar discontinuity, oxygen octahedral coupling between orthorhombic and cubic structures, intermixing, etc.)?*

In this case, we are interested in the metallic phase so we expect that phenomena associated to the presence of charged interfaces and the large induced electrical potential will not play a role here. Other effects such as oxygen octahedral coupling are expected to have an effect only for ultra-thin films but no gradual dependence is expected for large thicknesses.

-Q8. *In the text, it is mentioned that NNO/LAO grows under "small compressive strain", as one can derive from the lattice parameter difference of both compounds. However, in the abstract is mentioned "strain free". What is the conclusion?*

We agree it is confusing. The NNO films grown on LAO substrates are subjected to very small compressive strain, which is only -0.3 % but, indeed, not negligible. However, as shown in Figure 2b of this work, these films on LAO show the same $n = 1$ value as bulk NNO, which is exactly strain free. That is why consider that the $n = 1$ is the “strain-free” value. We have now slightly modified the abstract changing “Strain-free films...” by “Thin films under low strain conditions...”

-Q9. By the way, given that one wants to study the evolution of resistivity as function of strain, the strain values should be indicated.

Thanks for the suggestions. We have added a table with these values in the supplementary information.

-Q10. Fig. 1a: How do the authors explain the shift in tendency of the 7.5 and 10 nm NNO/LAO films? Has this result been reproduced?

Yes, it has been reproduced. Figure 7 (top), shows additional data of the TMI of two 7.5 nm thick films, two 10 nm thick films and one 8.5 nm thick film, which clearly reproduce the change of trend. So this behavior has been found to be robust in our films grown on LAO and we are currently investigating its origin, which we aim to discuss in a next paper.

The dependence of T_{MI} with thickness has been shown to be far from trivial (see e.g. the work of Peng et al. [1]) and our data show that the complexity could be larger than earlier thought: In Figure 7 (bottom), the T_{MI} trend is plotted for an extended temperature range. Neglecting, for the time being, the above-mentioned dip around 7.5nm, the T_{MI} seems to generally decrease for the lower thicknesses and to increase, towards the bulk value, at larger thicknesses. The later has also been reported in Ref. [1]. In this paper, based on their x-ray linear dichroism results, the authors convincingly propose that the changes at larger thickness are those expected as a result of strain relaxation and that at lower thicknesses the films are below the critical thickness for strain relaxation. In this “strained” regime the opposite T_{MI} trend is observed due to changes in orbital polarization: the degree of orbital polarization decreases for increasing thickness, with preference for the x^2-y^2 orbitals for the films on STO (tensile) and preference for the $3z^2-r^2$ orbitals for the compressive films on LAO. According to them the orbital degenerate state (like bulk) is reached for a

thickness in between 45 and 75 u.c., which would correspond to thicknesses between 20 to 30 nm, not far from the change of trend we also observe.

A possible explanation proposed for this low thickness effect is the gradient of defects across the film. However, no microscopic evidence is shown. In addition, these authors do not report a dip at 7.5-8.5 nm thickness. We believe it is important to clarify this behaviour. Our preliminary TEM data (see Figure 8) show a set of half integer reflections present in the 5 nm films; while these reflections are absent for thicker films. One possibility we are investigating is that the LAO substrate imprints its characteristic rotation pattern of AlO_6 octahedra in the NNO film up to 8 nm approx.

Since clarifying this behaviour requires more experiments and since we know that the different T_{MI} trend does not correlate with the change on the resistivity exponent n (the subject of the current paper), we believe that this discussion should be left out of the current paper.

Figure 7. Metal-insulator transition temperature (T_{MI}) of NNO/LAO films with different thickness.

Figure 8. HAADF-STEM images of (a) 5 nm and (b) 10 nm NNO/LAO films. (c) and (d) show the Fast Fourier Transform (FFT) analyses of (a) and (b) in the selected regions, respectively. The dotted red circles represent half-order spots, which usually come from oxygen octahedral rotations.

Reference:

[1] Peng, J. J., et al. "Manipulating the metal-to-insulator transition of NdNiO₃ films by orbital polarization." *Physical Review B* 93.23 (2016): 235102.

Minor comments:

-Q11. *Mention substrate orientation, especially for orthorhombic substrates.*

The LAO, STO and NGO substrates are <001> oriented, and the DSO substrates have the <110> orientation (this is to maintain the orthogonal in-plane lattice in this orthorhombic substrate). The orientation of the substrates have been added into the manuscript.

-Q12. *Indicate how strain is calculated.*

In the manuscript we mention the strain of the in-plane lattice parameters (a) of the films. For obtaining this, we perform a reciprocal space map scanning around (103)_c peak of each film, as shown in Fig. 3 in the manuscript. The value of $k/2k_0$ (in the units used in the manuscript) at the center of the film peak can be extracted and then the in-plane lattice parameter is calculated according to:

$$\frac{k}{2k_0} = \frac{\frac{2\pi}{a}}{2\frac{2\pi}{\lambda}}$$

where λ is the wavelength of the X-ray, which is 1.5406 Å in our system. The in-plane strain of the film is then calculated with respect to the unit cell volume of bulk NdNiO₃ as:

$$\varepsilon_{xx} = \frac{a - a_b}{a_b}$$

where $a_b = V^{1/3}$, being V the unit cell volume of bulk NdNiO₃.

Moreover, the in-plane strain of some films, such as 5 nm, 10 nm, 20 nm and 40 nm is in agreement with the geometrical phase analysis of STEM image of each film (Figures 4-5 of the manuscript).

-Q12. *X-ray diffraction pattern of the 40 nm NNO/LAO film: I believe there is a typo because it looks much thinner.*

Thanks for spotting this error. This is, indeed, the pattern of the 20 nm film. We have now corrected it.

-Q13. *Reference titles: ndnio3 should be replaced by NdNiO3*

We have corrected it.

Reviewer #2

This paper reports on the mechanism of the metallic conduction in perovskite nickelates as investigated by the analysis of the exponent of temperature-dependence of the resistivity. The conduction mechanism in nickelates is a long-standing issue of controversy. The authors prepared thin films of nickelate under different degrees of epitaxial strain. The resistivity of the film under small epitaxial strain shows T-linear dependence, reflecting the Fermi-liquid-like conduction. As increasing the epitaxial strain in the films, the temperature exponent becomes larger, indicating that the transport mechanism changes into non-Fermi-liquid-like conduction by the strain. The authors have investigated how the strain affects the carrier transport in the films by employing the TEM observation. It revealed that the quenched disorder induced by the strain dominates the change in the transport mechanism rather than the modification of the orbital degeneracy. The data shown in this paper is very clear and the deduced conclusion seems to be reasonable. This paper may contribute to converging the divergent opinions on the transport mechanism of metallic nickelate. The referee considers that the conclusion of this paper will be more strengthened by plotting the temperature exponent the resistivity as a function of the defect density rather than the strength of the epitaxial strain (Fig. 2b)..

We are grateful that the referee believes that this paper can contribute to clarify the controversies existing about metallic transport in nickelates. We agree that plot of exponents as a function of defect density will be more direct to clarify the effect of quenched disorder on the exponents. But the challenge is the extraction of defect density of films. Since oxygen vacancies would be associated to the presence of Ni^{2+} , a quantitative analysis $\text{Ni}^{3+} / \text{Ni}^{2+}$ ratio of

each film would be useful. We have performed XPS measurements on our samples (see figures 8-9) with that goal but the extraction of the $\text{Ni}^{3+} / \text{Ni}^{2+}$ is not possible due to the experimental uncertainty of (± 0.8 eV) of the equipment and the difficulty in peak fitting. To the best of our knowledge, this is also not been possible in other literature reports.

Transport data (Hall effect) is affected by the magnetic contribution, which precludes a direct correlation between Hall resistivity, charge density and oxygen vacancies.

Therefore, we keep the original plot of “n” vs strain, although we added a sentence in the revised version to clarify this point: *“Actually, to directly clarify the effect of disorder on n, a plot of n vs. defect density instead of epitaxial strain in Fig. 2b would be more appropriate. However, an accurate quantitative estimation of the amount of defects in such thin films is very challenging and could induce to erroneous conclusions (see Figure S7 in Supplementary information). Given the relationship between strain and defect concentration demonstrated by several authors [58,62,63], such a conservative plot is more adequate.”*

Reviewer #3

Nickelates have been attracting a great attention especially after the realization of the superconductivity in infinite layer nickelates. The authors timely provide deeper understanding of the MIT in nickelates and are important to the rapid growing field. After going through the manuscript in detail, I found couples of scientific questions which need to be addressed before I recommend the publication of the manuscript in Nature Communications.

We thank the referee for pointing out the importance of our work.

-Q1. First of all, a lot of structure evidence is shown for the deviation of the scaling exponent n from $n = 1$ and most of the discussion is also focused on tensile strain region, while I found very little discussion on the compressive part. What is the mechanism responding for compressive part should be explicitly addressed.

Following Patel et al., the tunable exponents of strongly correlated material can be achieved by the combined effect of Coulomb interaction (U) of 3d-electrons and the disorder (V) [1]. In nickelates, the Coulomb interaction is related to the orbital splitting of Ni 3d. The epitaxial strain lifts degeneracy and causes orbital polarization of the e_g band, which has two symmetries: $x^2 - y^2$ and $3z^2 - r^2$. As reported by *E. Mikheev et al.*, a larger compressive strain will increase the splitting by lowering the energy of $3z^2 - r^2$ orbitals, while tensile strain will lower the energy of the $x^2 - y^2$ orbital in a nearly symmetric fashion [2].

In this sense, both the compressive and tensile strain can have an influence on U . However, since we expect less defects for compressive strain, the observed n values for compressive strain films should be a closer measure of the direct effect of strain in the absence of disorder.

References:

[1] Patel, Niravkumar D., et al. "Non-Fermi liquid behavior and continuously Tunable resistivity exponents in the Anderson-Hubbard model at finite temperature." *Physical review letters* 119.8 (2017): 086601.

[2] Mikheev, Evgeny, et al. "Tuning bad metal and non-Fermi liquid behavior in a Mott material: Rare-earth nickelate thin films." *Science advances* 1.10 (2015): e1500797.

-Q2. The second comment is why the TMI decreases with increasing thickness is not explicitly addressed. It is expected to see that the TMI will move forward to bulk like for thicker films due to strain relaxation, which indicates the TMI should increase with increasing thickness for NNO/LAO. I would agree with the explanation for NNO/NGO where the strain relaxation picture well stands with the observed results. For ultrathin film, the interfacial modulation of the octahedral tilt modified by substrates should play a role, since the recent report does show the import role of interfacial oxygen octahedral coupling for nickelate when nickelate is very thin.

Thanks for the comment. We agree with the referee that the T_{MI} is expected to increase to bulk-like values for thicker films due to strain relaxation. In our work, we had also grown some thicker (> 40 nm) NNO/LAO films. The corresponding T_{MI} and sheet resistance are plotted in Figure 7. Indeed, we observe that for films above 40 nm, the T_{MI} increases with increasing thickness, which is consistent with the expectations. However, the reversed trend observed in films below 40 nm indicates that another effect different than strain relaxation is at play. This observation is consistent with that reported by Mikheev et al. [1] and by Peng et al. [2]. In this latter paper they assign the behaviour of the thinner thicknesses to gradual decrease in orbital polarization as thickness is increased and they propose this to originate in the defect depth profile. Next to these two regimes, we have also observed more complex behavior in the low thickness regime (as we also explained in our reply to referee 1), with a minimum T_{MI} at thicknesses close to about 8 nm. This is very reproducible and unexpected and we are investigating its origin in detail.

References:

[1] Mikheev, Evgeny, et al. "Tuning bad metal and non-Fermi liquid behavior in a Mott material: Rare-earth nickelate thin films." *Science advances* 1.10 (2015): e1500797.

[2] Peng, J. J., et al. "Manipulating the metal-to-insulator transition of NdNiO₃ films by orbital polarization." *Physical Review B* 93.23 (2016): 235102.

-Q3. Third and more technically, how to define T_{MI} is not clear, especially for MIT with large hysteresis, such as TMIT for 5 nm NNO/LAO.

The T_{MI} is defined by the temperature at which the slope of the resistivity versus temperature changes sign. All the T_{MI} we mention in Figure 1 and all the analysis of exponents are extracted from the resistivity data during the cooling down process. We have now added that in the main text and the caption of Figure 1.

-Q4. The last comment is that is there direct evidence to show the presence of the Ni²⁺ in tensile nickelate, such as EELS or XAS? This will strongly validate the role of oxygen vacancies and the proposed model.

Thanks for the suggestions, which we consider for future work. We have performed XPS measurement on the NNO/STO films, which shows a clear Ni 2p_{3/2} peak. After taking the Au 4f_{7/2} peak into account, the position of the Ni 2p_{3/2} peak is observed to be 854.86 eV, which is located in between the expected Ni³⁺ peak (855.9 eV) and the Ni²⁺ peak (853.9 eV). The fitting to the peak indicates the coexistence of Ni³⁺ peak and Ni²⁺ peak. However, due to the experimental accuracy of the measurements (± 0.8 eV) and the necessarily simplified fitting (e.g the XPS spectra of Ni³⁺ and Ni²⁺ peak should be considered

as a sum of six peaks [1]), the extraction of an exact ratio of $\text{Ni}^{3+} / \text{Ni}^{2+}$ is impossible for us.

In addition, the films on LAO also show coexistence of Ni^{2+} and Ni^{3+} peaks. This has also been reported by Preziosi et al. [1], which showed that even the samples with the sharpest MIT and resistance changes of nearly 6 orders of magnitude still showed a $\text{Ni}^{2+}/\text{Ni}^{3+}$ ratio of about 0.2. This makes the quantification of the relative Ni^{2+} in the different samples very challenging.

Figure 8. Ni 2p XPS spectrum of 5 nm (a) NNO/STO film and (b) NNO/LAO film.

Figure 9. Comparison of Ni 2p XPS spectrums between NNO/STO and NNO/LAO films.

Reference:

[1] Preziosi, Daniele, et al. Reproducibility and off-stoichiometry issues in nickelate thin films grown by pulsed laser deposition. *AIP Advances* 7.1 (2017): 015210.

Reviewers' comments:

Reviewer #1 (Remarks to the Author):

Despite the manuscript presents an extensive characterization of the NdNiO₃ thin films, after the reply of the authors I can still unfortunately not recommend the paper for publication to Nature Communications given that the paper does not represent the important advances of significance to the field that the journal aims to. The reply of the authors confirms that the conclusions of the manuscript might depend on the growth conditions used. For example, the main conclusion is that tensile strain favors disorder. Growth conditions have only been optimized for films grown on LAO (nearly almost free), and thus it is not clear how conclusions about influence of strain/disorder will change if optimization was also carried out for films grown on STO (tensile strain). Furthermore, enhanced formation of oxygen vacancies as tensile-strain increases is a rather well-known phenomenon on oxides. The presence of Ruddlesden-Popper faults is another indication of non-stoichiometric films, even accordingly to the paper cited by the authors.

Reviewer #2 (Remarks to the Author):

The authors have responded to the reviewer's comments. The revised manuscript can be published without any further correction.

Reviewer #3 (Remarks to the Author):

In the revised manuscript the authors have addressed my criticism regarding the effect of compressive strain, and thickness effect on MIT and the other evidence of oxygen vacancy. Given that the Ni²⁺ is also present in NNO/LAO as the authors discussed in a cited reference (e.g., Ni²⁺/Ni³⁺ ration of about 0.2), this may weaken the conclusion of oxygen vacancy dominated the change of transport index. Presence of Ni²⁺ no matter what indicates the appearance of defects. This is not consistent with the claim that compressive strain film has much less defects. Therefore, I still have concern about the oxygen vacancy model. I would suggest a more solid argument/evidence on this, with which I then will be able to support the publication of the manuscript in Nature communications.

Although we cannot offer to publish your paper in Nature Communications, the work may be appropriate for another journal in the Nature Research portfolio. If you wish to explore suitable journals and transfer your manuscript to a journal of your choice, please use our <https://mts-ncomms.nature.com/cgi-bin/main.plex?el=A6S5Benc1B3aKe1X1A9ftdPfHFxgynshPK1rDciiyQAZ> manuscript transfer portal. If you transfer to Nature-branded journals or to the Communications journals, you will not have to re-supply manuscript metadata and files. This link can only be used once and remains active until used.

All Nature Research journals are editorially independent, and the decision to consider your manuscript will be taken by their own editorial staff. For more information, please see our http://www.nature.com/authors/author_resources/transfer_manuscripts.html?WT.mc_id=EMI_NPG_1511_AUTHORTRANSF&WT.ec_id=AUTHOR manuscript transfer FAQ page.

Appeal NCOMMS-19-30498A

Point-by-point reply to Reviewers' comments:

Reviewer #1

Despite the manuscript presents an extensive characterization of the NdNiO₃ thin films, after the reply of the authors I can still unfortunately not recommend the paper for publication to Nature Communications given that the paper does not represent the important advances of significance to the field that the journal aims to.

First, we like to emphasize that most of the papers on nickelates have focussed on the metal-insulator transitions and there are relatively less investigating the character of the metallic phase. The conclusions of our paper fully align with (and are the first experimental demonstration of) the theoretical predictions on Patel, Dagotto *et al.*, which were considered important enough to be published in *Phys. Rev. Lett.* [PRL 119, 086601 (2017)] even then. More recently, the interest in the metallic phase of nickelates has been revived due to the discovery of superconductivity in infinite-layer nickelates (Li *et al.*, *Nature* 572, 624–627 (2019)). In particular, we report the effect of disorder on the metallic phase of NdNiO₃, the parent phase of superconducting Nd_{0.8}Sr_{0.2}NiO₂ reported in the *Nature* paper. Therefore, we are convinced that the results reported in our manuscript have become even more significant, and that *Nature Communications* is the best forum for our paper to reach a wide audience.

“The reply of the authors confirms that the conclusions of the manuscript might depend on the growth conditions used.”

This is an incorrect interpretation of the conclusions of the paper. Several papers in the literature found a temperature exponent of the resistivity $n = 2$ or $n = 5/3$, which is commonly associated to non-Fermi liquid physics and proximity to quantum critical point. This is a hot topic in oxides, in general, and in Nickel oxides and NdNiO₃ in particular after the discovery of superconductivity in a phase derived from this material.

In our paper, we carefully change the growth conditions to tune the concentration of oxygen vacancies in our films and we demonstrate that the value of the exponent n depend very much on those growth conditions. Therefore, we show that the exponent n may take different values from $n = 2$ to $5/3$ and beyond, if a large enough concentration of defects is present in the sample, without invoking exotic physics.

In short, the results of the experiments depend on the growth conditions (because those are our control parameters), not the conclusions of the paper. This is an important difference, which the referee did not seem to appreciate.

In any case, the possibility of tuning the resistivity exponent with quenched disorder is a very important conclusion, particularly timely at this time, where the nature of the metallic phase in nickelates is being extensively discussed, in view of the latest results showing superconductivity in an infinite-layer nickelates.

“For example, the main conclusion is that tensile strain favors disorder.”

This statement is incorrect. Honestly, we do not understand where the referee gets this conclusion from, which is not identified as such in the abstract or in the conclusions of our paper. It is a well-known fact that oxygen vacancies are favoured under tensile strain in 3d metal oxides: ionized O²⁻ vacancies donate two electrons to antibonding Ni-O states, which therefore expand and accommodate to the epitaxial tensile stress induced by the substrate.

We cite several papers, which corroborate this effect, and we added more references, upon request of the referees. We clarified already repeatedly in our reply that we are not claiming that this is an original result (neither a conclusion of our paper). Honestly, we do not understand why the referee insists on this point.

“Growth conditions have only been optimized for films grown on LAO (nearly almost free), and thus it is not clear how conclusions about influence of strain/disorder will change if optimization was also carried out for films grown on STO (tensile strain).”

Indeed, we optimized the growth condition for NdNiO₃ during the growth on LAO and then we used the same optimized conditions to grow the same materials on the other substrates. In this way we fix all growth conditions except strain, which is changed with respect to the reference substrate (LAO). We use this approach because it is the only way to change just one parameter (strain) at a time. Referees #2 and #3 understood this properly.

It would be possible to modify the growth conditions from sample to sample, to generate samples on STO with “optimized resistivity”. For example, a larger oxygen pressure would (partially) compensate the oxygen vacancies created due to the additional tensile strain. But this strategy would prevent a systematic study of the effect of oxygen vacancies on the resistivity, which is precisely the goal of the paper.

So, honestly, we are puzzled by this comment by Referee #1

“Furthermore, enhanced formation of oxygen vacancies as tensile-strain increases is a rather well-known phenomenon on oxides”

We agree with this comment, and we say that in our paper and in our previous reply. As mentioned above, this is not a result from our paper; it is just a statement that helps strengthening our argument that films on STO present more vacancies than films on LAO. We do not understand why the reviewer insists on writing that we claim this as novel.

“The presence of Ruddlesden-Popper faults is another indication of non-stoichiometric films, even accordingly to the paper cited by the authors.”

Yes, we agree and also mentioned that several times in the paper and in the reply to the referees. All previous reports on nickelates that include a detailed microscopy analysis have shown the presence of R-P faults. In our thinnest films deposited on LAO (under optimized growth conditions) these defects are very scarce, while they are more abundant in the other films.

However, if we look at the resistivity, the 6-7 order of magnitude change of resistivity at the metal-insulator transition and the very low resistivity values in the metallic phase, are a strong evidence of the high quality of our LAO films. Note that transport properties probe the bulk properties of the film (contrary to local probes, such as TEM), and therefore the resistivity data place our films among the best reported in the literature so far. This was well motivated in the reply to the referees.

Thus, we are not claiming that the films are defect-free, this is simply impossible. We have enough experience on thin film growth of complex oxides to know that a fully stoichiometric film is not attainable. This is especially hard in nickelates, for which it is well known that stabilizing the bulk perovskite phase needs of high pressures. However, we have experimental data to demonstrate that the films are among the best in the literature in terms of RP faults, intergrowths and any other type of unavoidable extended defects.

Reviewer #2:

The authors have responded to the reviewer’s comments. The revised manuscript can be published without any further correction.

We are glad that the Reviewer has properly understood the importance of the results and the adequacy of the experiments to support them.

Reviewer #3 :

In the revised manuscript the authors have addressed my criticism regarding the effect of compressive strain, and thickness effect on MIT and the other evidence of oxygen vacancy. Given that the Ni²⁺ is also present in NNO/LAO as the authors discussed in a cited reference (e.g., Ni²⁺/Ni³⁺ ration of about 0.2), this may weaken the conclusion of oxygen vacancy dominated the change of transport index. Presence of Ni²⁺ no matter what indicates the

appearance of defects. This is not consistent with the claim that compressive strain film has much less defects. Therefore, I still have concern about the oxygen vacancy model. I would suggest a more solid argument/evidence on this, with which I then will be able to support the publication of the manuscript in Nature communications.

Our experience with the growth of complex oxide thin films makes us very much aware that obtaining vacancy-free or fully stoichiometric thin films, as well as the characterization of the vacancy content, is very challenging. In nickelates that problem is even greater as, even in bulk, high pressure is needed to achieve the perovskite phase ; the small $p-d$ charge transfer energy in Ni^{3+} RNiO_3 perovskites, stabilizes a hole in the ligand $2p$ band, making these systems metastable). That is why we have not claimed that the samples on LAO are vacancy-free. Indeed, in the manuscript we write “*Since the amount of defects is smaller in the films under compressive strain, the values of n under epitaxial compression should be a closer measure of the direct effect of strain in the absence of disorder.*”

However, we are happy to include some more recent experiments showing that the amount of defects is strongly reduced in the films on LAO. For that, we have added the following text just before the penultimate paragraph (see red text in the suitable version attached):

“In addition, a direct investigation of the correlation between electrical transport properties and defect density can be achieved by tuning the concentration of oxygen vacancies of a single film by changing the annealing conditions after growth. For this, a 20 nm NNO film grown on a STO substrate with different amounts of oxygen vacancies was prepared in this work (see methods and section 6 in SI) and the corresponding changes in structure and resistivity were characterized (see Fig. S6). As we mentioned above, the existence of oxygen vacancies gives rise to an enlarged unit cell volume of the films. This is an effect of chemical expansivity due to electrons being donated to σ bands. Hence, the change in the density of oxygen vacancies is correlated with a change of the lattice parameters of the films. As shown in Fig. S6 (a), the out-of-plane lattice parameter of the 20 nm NNO/STO film after vacuum annealing is about 3.799 Å. This value is larger than the 3.782 Å of the optimized film (see Fig. 3 (b)), which has been annealed with a 900 mbar oxygen pressure, as explained in the Methods. This is consistent with a larger content of oxygen vacancies for the vacuum-annealed films, as expected. As a consequence of this increase of oxygen vacancies, the metallic phase is fully suppressed, accompanied with several orders of magnitude increase in resistivity, as shown in Fig. S6 (c). If the film is subsequently annealed in an oxygen-enriched environment at increasingly large temperatures, the oxygen can be gradually replenished, resulting in a decrease of the out-of-plane lattice parameter and, thus, a shift of (002) diffraction peak toward larger angles. Correspondingly, the resistivity shows a decrease and the metallic phase is recovered after annealing at sufficiently high temperature. More importantly, with the further reduction of oxygen vacancies, a clear evolution of the exponent n from 2.24 to 1.64 is also observed in the resistivity of the metallic phase (see inset in Fig. S6 (c) and Fig. S7), deviating from the $T^{1.33}$ dependence measured for this thickness on samples annealed with the standard procedure (see Fig. 2 (a)). For comparison, the same annealing treatment was also employed in a 20 nm NNO/LAO film. However, only a linear- T dependence of resistivity ($n=1$) is found in this system after the recovery of the metallic phase, regardless of the oxygen content (see Fig. S6 (c) and (d)). These experiments reveal that the oxygen vacancy content in the films on LAO is not large enough to induce changes in the macroscopic transport through the film, while the larger oxygen vacancy content in tensile-strained nickelate films clearly affects the resistivity-temperature scaling exponent.

Next to vacuum-annealing, large enough tensile strain can also induce a large density of oxygen vacancies and should, eventually, suppress the metallic phase....”

In addition, related to the paragraph above, we have added a new section in the supplementary information (section 5), with two new figures (see new version attached).

REVIEWERS' COMMENTS:

Reviewer #3 (Remarks to the Author):

In revised manuscript, the authors have addressed my concerns and I recommend the publication of the manuscript.